# Efficient Network Structure via Ensemble Deep Evolutionary Synthesis

**Keyvan Kasiri, Mohammad Javad Shafiee, Alexander Wong**
Vision and Image Processing (VIP) Research Group
Systems Design Engineering
University of Waterloo
Waterloo, ON N2M, Canada
`{kkasiri, mjshafiee, awong}@uwaterloo.ca`

**Justin Eichel**
Miovision Technologies Inc.
Waterloo, ON, Canada
`jeichel@miovision.com`

## Abstract

While deep neural networks have shown promising results in a wide range of applications on highly powerful computational devices, one challenging task is to deploy a deep neural network on embedded devices for the widespread use. Deep neural networks and specially convolutional neural networks are usually over-parameterized and one possible solution is to remodel the network architecture with a smaller network architecture with a trade-off on modeling accuracy and performance. Here we take advantage of meta-learning algorithms to synthesize a more efficient model while it boosts the modeling performance. To this end, we propose an ensemble of deep evolutionary intelligence frameworks where it synthesizes several very efficient models with less than 3% drop on modeling accuracy and then aggregates them to boost the modeling performance. Experimental results demonstrates that the proposed ensemble of Deep Evolutionary Synthesis approach synthesizes an ensemble model which is 1.5X smaller than the original network architecture while performing more accurate (83.30% compared to 83.18%) than the original network in terms of modeling accuracy for binary object segmentation.

## 1 Introduction

Deep neural networks (DNNs), specifically convolutional neural networks (CNNs), have made significant progress in a wide range of applications across a number of domains LeCun et al. (2015); Bengio et al. (2009). However, due to the large number of parameters, deep learning methods often require significant amount of memory and computational resources. Despite the fact that new deep learning solutions have offered promising detection and classification performance, deployable real-time methods are still strongly required, particularly for embedded applications. Structural redundancy of traditional deep neural networks leads to increasing the required memory as well as training and decision time, which intensifies the deployment cost of such methods in embedded platforms Gong et al. (2014); Liu et al. (2015).

Recently, a wide range of approaches have targeted deep neural networks to reduce the redundancy in parameters of the deep networks, and therefore attain efficient networks in terms of computations and required memory. Among them, one can refer to compressing deep neural networks using vector quantization Gong et al. (2014), pruning to lower the network complexity as well as over-fitting Hanson & Pratt (1989); LeCun et al. (1990), combining quantization and pruning Han et al. (2015), and evolutionary synthesis of deep models Shafiee & Wong (2016); Shafiee et al. (2016) as some examples. However, reducing the redundancy in deep neural networks, in general, is prone to having loss in detection accuracy.

Ensemble learning has been traditionally used as an effective solution to attain a model with improved generalisation ability Dietterich (2000; 2002). Recently, ensemble methods have been employed in a number of applications to combine different variations of deep neural networks, and therefore improve the prediction performance Ciregan et al. (2012); Deng & Platt (2014); Du et al. (2017). Multi-Column deep neural network was proposed in Ciregan et al. (2012) to boost recognition performance for traffic sign classification by combining various DNNs trained on differently preprocessed data.

In this paper, an efficient architecture of deep neural network is presented based on ensemble of highly sparsified variations of a model. Model sparsification is performed by taking advantage of the evolutionary deep intelligence framework Shafiee & Wong (2016) to synthesise highly efficient independent networks. Over the evolutionary procedure, each network is trained towards yielding a highly sparse set of synaptic weights and clusters across successive generations of evolution. The ensemble of highly sparse neural networks is desired to achieve a high detection performance, yet maintaining the capability to reduce the deployment costs.

## 2 METHODOLOGY

In this paper, the objective is to achieve an efficient architecture for a deep neural network model suitable for embedded applications, while improving the detection accuracy of the model. The proposed framework works on the basis of ensemble learning of generated models through the evolutionary synthesis.

**Evolutionary Framework.** The evolutionary synthesis of deep neural networks (Evo-net), inspiring from biological evolution, tries to achieve an efficient architecture for the original deep model Shafiee & Wong (2016). In evolutionary approach, the deep neural network model evolves in successive generations to yield highly efficient networks. This approach aims to mimic natural selection, heredity, and random mutation from biological evolution. The architectural evolution is formulated based on computational environmental factors in a random manner and a synaptic probability model, where descendant network is synthesized relying on these synaptic probability models from the ancestor network.

The genetic framework for the deep neural network architecture $\mathcal{H}$ with a set of possible synapses $S$ and a set of the synaptic strength $\mathcal{W}$ is formulated as a conditional probability of the architecture in synthesised in generation $g$ given the architecture of its ancestor in generation $g-1$,

$$P(\mathcal{H}_g) = \mathcal{F}(\mathcal{E}) \cdot P(S_g|\mathcal{W}_{g-1}), \tag{1}$$

where $\mathcal{F}(\mathcal{E})$ models the environmental factor to computationally limit resources available for the descendant networks. The term $\mathcal{F}(\mathcal{E})$ constrains the number of synapses that can be synthesized in the descendant network and is set to $\mathcal{F}(\mathcal{E}) = K$, where the quantity $K$ enforces the highest percentage of synapses desired in the descendant network.

To have a more efficient encoding scheme, synaptic clustering was proposed in **?** to improve the demands for memory and storage, as well as adaptability for parallel computations in GPUs. The synthesis procedure in Eq. 2 is reformulated as

$$P(\mathcal{H}_g) = \prod_{c \in C}[\mathcal{F}_c(\mathcal{E})P(\bar{s}_{g,c}|\mathcal{W}_{g-1}) \cdot \prod_{i \in c}\mathcal{F}_s(\mathcal{E})P(s_{g,i}|w_{g-1,i})], \tag{2}$$

where $\mathcal{F}_c(\cdot)$ and $\mathcal{F}_s(\cdot)$ stand for the environmental factors enforced at the cluster and synapse levels, respectively. Here $s_{g,c} \in S_g$ and $\bar{s}_{g,c} \subset S_g$ denote a particular synapse and a particular cluster of synapses at generation $g$ for a given cluster $c$, and $w_{g-1,i} \in W_{g-1}$. The specific synaptic cluster in a deep convolutional architecture is allowed to be any subset of synapses such as a kernel or a set of kernels. The descendant networks are synthesised and trained, and the evolutionary synthesis process is successively repeated to achieve descendant networks of desired characteristics.

**Ensemble of Classifiers** Bagging Breiman (1996), as a popular and effective ensemble method, aims to increase the stability of learning algorithms. The ensemble of networks in this paper, follows bagging scheme, which is simply formed by taking average of the output of activations from several networks generated through the evolutionary synthesis. For a given input image, the prediction of all synthesised networks contributing the in ensemble are averaged.

Table 1: Synaptic and cluster efficiency, A-E$_1$ and A-E$_2$, vs detection accuracy, $F_\beta$ at the forth generation of the synthesized network.

| Gen | A-E$_1$ | A-E$_2$ | $F_\beta$ |
|---|---|---|---|
| 0 | 1 | 1 | 0.8318 |
| 1 | 1.17 | 1.12 | 0.8294 |
| 2 | 1.42 | 1.28 | 0.8273 |
| 3 | 2.92 | 1.76 | 0.8136 |
| 4 | 4.57 | 2.14 | 0.8082 |

Table 2: Detection accuracy,$F_\beta$, of the original network, synthesised networks, M$_1$, M$_2$, M$_3$, and ensemble of networks.

| Original | M$_1$ | M$_2$ | M$_3$ | Ensemble |
|---|---|---|---|---|
| 0.8318 | 0.8182 | 0.8201 | 0.8250 | 0.8330 |

## 3 EXPERIMENTAL RESULTS AND DISCUSSION

In order to assess the efficacy of the proposed framework, the evolutionary synthesis is performed across several generations, over parallel branches, to generate sparse deep models. The network architecture efficiency and accuracy is assessed using MSRA-B dataset Jiang et al. (2013). The MSRA-B dataset contains 5000 images, and is highly used for the application of visual saliency detection. For the experiments, the dataset is divided into training, validation, and test subdivisions, each containing 2500, 500, 2000 images respectively. The training and validation sets are augmented using horizontal flipping for the training purpose. The model proposed in Luo et al. (2017), as a high performance saliency detection method based on a multiscale structure, is opted as the original network architecture. The network settings are set to the values suggested by the authors in the original model Luo et al. (2017).

The deep neural network with the above setting is trained over a number of generations. At every single generation, the model is forced by the environment factors to randomly remove a percentage of the parameters. Table 1 illustrated the synaptic and cluster efficiency of the synthesised at the forth generation of the synthesized deep neural network. As is shown, the evolutionary framework is capable of achieving highly efficient models, in terms of synaptic and cluster efficiency, with less than 3% drop in detection accuracy. The evolutionary framework is performed to synthesize three different variations from the original network model. The ensemble of synthesised networks is formed to perform the saliency detection over the test dataset. Table 2 refers to the detection accuracy results from the synthesised networks in four generations, illustrated by M$_1$, M$_2$, and M$_3$, and the ensemble results. Detection accuracy using ensemble of networks takes advantage of variations in three contributing networks and could improve the accuracy compared to the original network. While, achieving an improvement in accuracy, the ensemble of network still benefits from the smaller number of parameters compared to original network. Based on the quantitative results, applying Evo-Net in an ensemble framework significantly improves the architecture efficiency of the model in formation of highly sparse synaptic weights and clusters, and therefore facilitates the adaptation for highly parallel computations such as GPUs. Yielding an efficient, yet powerful deep models for vehicular applications can lead to a promising direction for future exploration in embedded deep learning.

## ACKNOWLEDGMENTS

This research is funded by the SOSCIP TalentEdge Post-doctoral Fellowship Program in partnership with Ontario Centres of Excellence (OCE), and supported by Miovision Technologies Inc.

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
