# OpenReview forum: "Efficient Network Structure via Ensemble Deep Evolutionary Synthesis"
_ICLR.cc/2018/Workshop — Reject_

### Official Review · AnonReviewer3 · 2018-03-07
**The idea of generating neural networks using evolutionary computing is not new**

**Rating:** 5
**Confidence:** 5

**Review:**

The authors of this paper propose an idea of generating CNN networks using evolutionary computing. Generally speaking, this idea has a long history, where the main challenges are coding strategies and computational efficiency. This paper does not answer the essential questions in this topic: How many generations are needed for the producing good individuals? In each generation, what the number of individual networks? How many networks are sufficient in the ensemble learning based decision making?

Minors:
performing more accurate -> performing more accurately

Multi-Column deep neural network -> Multi-column deep neural network

inspiring from biological evolution -> inspired from biological evolution

was proposed in ?: missing citation

I could not understand this phase "in synthesised in generation g"

---

### Official Review · AnonReviewer1 · 2018-03-09
**Hard to understand, very few experiments.**

**Rating:** 2
**Confidence:** 4

**Review:**

As I understand it, this paper proposes a way to "evolve" a network into a set of sparse versions of the model; this evolutionary process is then used to "synthesize" several variations of the original model, which are then combined as an ensemble.  They show through an experiment on a small image dataset that the performance of the ensemble is able to achieve slightly better performance than the original model, even though it uses less memory.

My largest complaints about this paper are in regard to clarity, and quality of experiments.  The evolutionary process is barely described (the different terms in equations (1) and (2) are not defined precisely).  The writing is quite hard to understand.  The dataset used is rather small, and I was not familiar with it in advance.  Very few experimental results are reported; they simply synthesize three version of the same model, and take the ensemble.  They do not compare the results to baselines, such as other methods for network pruning, or consider using structured matrices in order to reduce the memory footprint of their models (their stated goal).  It is unclear to me what the novel contribution of this paper is, or whether the experimental results are significant (since they don't compare to baselines, and they run a very small number of experiments).

I am not familiar with the existing work on evolutionary synthesis of DNNs, so it is hard for me to comment on how this work builds on that work.

---

### Decision · Program_Chairs · 2018-03-20
**ICLR 2018 Workshop Acceptance Decision**

**Decision:**

Reject

**Comment:**

Based on the reviews, this paper has not been accepted for presentation at the ICLR workshop. However, the conversation and updates can continue to appear here on OpenReview.